# Epidemiology of Dementia in China in 2010–2020: A Systematic Review and Meta-Analysis

**DOI:** 10.3390/healthcare12030334

**Published:** 2024-01-28

**Authors:** Yueheng Yin, Hon Lon Tam, Jennifer Quint, Mengyun Chen, Rong Ding, Xiubin Zhang

**Affiliations:** 1School of Nursing, Nanjing Medical University, Nanjing 210029, China; yinyueheng@njmu.edu.cn; 2The Nethersole School of Nursing, Faculty of Medicine, The Chinese University of Hong Kong, Hong Kong 999077, China; hltam@cuhk.edu.hk; 3School of Public Health, National Heart and Lung Institute, Imperial College London, London W12 7RQ, UK; j.quint@imperial.ac.uk (J.Q.); r.ding23@imperial.ac.uk (R.D.); 4School of Nursing, Lanzhou University, Lanzhou 730000, China; chenmy20@lzu.edu.cn

**Keywords:** epidemiology, dementia, prevalence, incidence, mortality, China, systematic review, meta-analysis

## Abstract

Background: Dementia has become one of the leading causes of death across the world. Aims: The aim of this study was to investigate the incidence, prevalence, and mortality of dementia in China between 2010 and 2020, and to investigate any geographical, age, and sex differences in the prevalence and incidence of dementia. Methods: Five databases were searched. The Joanna Briggs Institute (JBI) critical appraisal tool was used to assess the quality of the included studies. A random-effects meta-analysis was performed to estimate the pooled prevalence of dementia. Subgroup analysis was based on the type of dementia. The incidence and mortality of dementia were synthesized qualitatively. Results: A total of 19 studies were included. The meta-analysis showed that the prevalence of dementia was 6% (95%CI 5%, 8%), the prevalence of Alzheimer’s disease (AD) was 5% (95%CI 4%, 6%), and the prevalence of vascular dementia (VaD) was 1% (95%CI 0%, 2%). The subgroup analysis showed that the prevalence rates of dementia in rural (6%, 95%CI 4%, 8%) and urban areas were similar (6%, 95%CI 4%, 8%). Deaths due to dementia increased over time. Conclusion: The prevalence, incidence, and mortality of dementia increased with age and over time. Applying consistent criteria to the diagnosis of cognitive impairment and dementia is necessary to help with disease monitoring. Promoting dementia knowledge and awareness at the community level is necessary.

## 1. Research Manuscript Sections

### Background

Dementia is an umbrella term for a variety of brain diseases that gradually deteriorate an individual’s memory, language, and other intellectual abilities [1]. It is recognized as a global priority due to the burden on public health, global economy, and the wider society. It has become the seventh leading cause of death across the world [2]. Most types of dementia cannot be cured, but there are ways to manage dementia symptoms and improve a person’s quality of life. Studies have indicated that an early and accurate diagnosis of dementia is important in the implementation of clinical interventions and treatments [3,4,5,6]. However, evidence shows that in lower- and middle-income countries (LMICs), 80% of people with dementia have not been diagnosed, which prevents them from accessing medical services and support [7]. In many countries, especially LMICs, there is a lack of epidemiological data on dementia due to the low rate of diagnosis [8]. Therefore, obtaining data on the epidemiology of dementia is important as it can be used to inform policy makers and health professionals on how to improve early diagnosis, as well as the development of dementia support services and related interventions.

A World Health Organization (WHO)’s report, ‘Global status report on the public health response to dementia’, shows that there were 55.2 million people worldwide living with dementia, with 60% of cases in LMICs, in 2021, and the expected prevalence will go up to 78 million in 2030 and 139 million in 2050 [2]. The economic costs of dementia are huge, and it is estimated that the cost of dementia was USD 1.3 trillion worldwide in 2019 [2]. China has the largest population of people with dementia in the world. However, it lacks updated national/large population-based epidemiolocal statistics on the prevalence of dementia. The World Health Organization utilized data from Chan, Wang [9], with the estimated number of people with dementia in China being 9.18 million in 2010. The existing estimated prevalence rates between studies and within regions are heterogeneous [10]. The official demographic statistics in China do not provide sufficient information. For example, statistics reporting the number of people with dementia are not available in the *National Statistical Yearbooks* and on the China Association for Alzheimer’s Disease website. Some regional population-based studies found that 7.7% of the population over 60 years had dementia in northern rural China [11], while that figure was 1.7 times higher (13%) in the Zhejiang province in southern urban China [12]. These diverse results could be related to the inclusion criteria or the different sampling methods used by these studies.

The WHO states that until recently, only 50 out of 194 WHO members have a national dementia plan, and so the WHO urges that more than 75% of countries should develop a national dementia strategy/plan by 2025 (WHO, 2021). Without clear figures on the incidence and prevalence of dementia, it is impossible to develop an accurate national dementia plan. Therefore, with regard to China’s prevalence of dementia, political, clinical, and social research is urgently needed to inform the government as well as health services of the importance of implementing evidence-based regulations, treatments, interventions, and care. In a previous systematic review, Chan, Wang [9] reported that the number of people with some form of dementia increased from 3.68 million in 1990 to 5.62 million in 2000 and then to 9.19 million in 2010. The prevalence of dementia may change over time due to a number of factors, including the development of testing techniques, the society’s emphasis on awareness and diagnosis, and changes in the age structure of the population [13,14,15]. Therefore, we conducted a systematic review of epidemiological studies of dementia in both Chinese and English from 2010 to 2020. We estimated the prevalence, incidence, and mortality associated with dementia and its subtypes in China. In addition, due to the large diversity of healthcare services, customs, and lifestyles across the country, differences in prevalence by age, sex, and geography were also investigated to inform related stakeholders For example, a previous study indicated that there was a greater inequity of outpatient service use in urban areas than in rural areas in China [16]. 

## 2. Methods

This systematic review was reported by following the PRISMA statement [17] and its checklist (please see Appendix A). The protocol of this review had been registered in PROSPERO (CRD42021249995).

### 2.1. Objective

In this review, we aimed to (1) investigate the incidence, prevalence, and mortality of dementia in China and any changes over time between 2010 and 2020, and (2) investigate any geographical, age, and sex differences in the prevalence and incidence of dementia in China.

### 2.2. Eligibility Criteria

The inclusion criteria were as follows: (1) population-based original observational studies; (3) studies reported the incidence and/or prevalence of any type of dementia; (4) studies conducted between 2010 and 2020; (5) studies conducted in mainland China; and (6) studies written in English or Chinese.

The exclusion criteria were as follows: (1) studies with no numerical estimates; (2) any type of review papers or conference abstracts; (3) studies without a comprehensive diagnostic assessment or did not use internationally recognized definitions of Alzheimer’s disease (AD) and other types of dementia; (4) studies with an incorrect use of multistage design, e.g., the sample size was too small or only the individual circumstances of specific subgroups or cases were analyzed; (5) studies with animals; (6) studies assessing males/females only, without targeting the whole population; and (7) study settings in nursing homes/hospitals.

### 2.3. Search Strategy

Several databases, including the China National Knowledge Infrastructure (CNKI) Wanfang, PubMed, Embase, and Scopus, were searched. The search terms are shown in Appendix A, which were combined with Boolean operators (OR/AND). A manual search was also performed on the reference lists of the obtained articles. Alerts of updated literature in the databases were set to receive emails monthly in order to avoid missing updates. The search was continued till May 2022.

### 2.4. Study Selection

Endnote X9 was used to manage the identified studies. Duplicates were first removed. Then, two reviewers independently screened the remaining studies in two steps: (1) preliminary screening of titles and abstracts, and (2) secondary screening of full-text articles according to the eligibility criteria. Any disagreements were discussed with a third reviewer to achieve a consensus. 

### 2.5. Data Extraction

Two reviewers extracted the data independently, and the third reviewer checked the accuracy. The information extracted from the articles included authors, year, study design, settings, sample size, population demographic information (age and gender), method of dementia ascertainment, prevalence value, incidence value, and mortality value.

### 2.6. Risk of Bias in Individual Studies

The risk of bias in the included studies was assessed by two reviewers independently by using the critical appraisal tool for prevalence studies from the Joanna Briggs Institute (JBI) [18]. The JBI critical appraisal tool contains nine items, with four rating choices (i.e., yes, no, unclear, and not applicable) for each item (details of the items are shown in Table 1). Any disagreements were resolved by discussion with a third reviewer to reach a consensus.

### 2.7. Data Analysis

A random-effects meta-analysis was performed using Stata 16.0 (StataCorp LLC, College Station, TX, USA) to estimate the pooled prevalence of dementia. Subgroup analysis based on the types of dementia (i.e., AD and VaD) was performed. In addition, subgroup analysis based on different areas and years (before and after 2015) in the prevalence of dementia was conducted. Statistical significance was defined as *p* < 0.05. Heterogeneity among the included studies was tested using the chi-squared test (significant if *p* < 0.10). Statistical heterogeneity was evaluated according to the I^2^ statistic by defining low heterogeneity as I^2^ < 25%, moderate heterogeneity as I^2^ equal to 25–75%, and high heterogeneity as I^2^ > 75% [36]. Descriptive analysis was used to present the incidence and mortality of dementia, as well as geographical, age, and sex differences in the incidence, prevalence, and mortality.

## 3. Results

### 3.1. Study Selection

A PRISMA flow chart showing the study screening process is shown in Figure 1. A total of 2630 records were identified from the database search, and three records were identified from a search of the reference lists of the relevant studies. After removing duplicates and screening the titles and abstracts, 120 records were left for full-text screening. Finally, 19 studies were included, the full list of which is shown in Appendix A.

### 3.2. Study Characteristics

The characteristics of the 19 included studies are presented in Table 2. Nine studies recruited participants from urban and rural areas, seven recruited participants from urban areas, and three recruited participants from rural areas. Bo, Wan [19] was the only study that assessed dementia cases using national datasets. Jia, Du [28] included data from 12 provinces, and the rest of the studies used regional data in terms of cities and provinces. The sample size of the included studies ranged from 129 to 77 million. Regarding age, Bo, Wan [19] included people of all ages, while Yu, Zhang [35] included people aged more than 80 years. In other studies, 12 studies included people aged 60 and above, three studies included people aged 55 and over, and two studies included people aged 65 and above. Five studies [19,22,25,34,35] adopted a longitudinal study design, and the rest were all cross-sectional studies.

The Mini-Mental State Examination (MMSE) was commonly used to diagnose dementia cases among the included studies [11,12,20,21,22,23,27,28,29,31,32,33,34,35], but the cut-off for dementia was slightly different among the studies. For example, Ji, Shi [11] used a score of 18 as the cut-off score of dementia for illiterate individuals, whereas Huang Fu, Chang [26] and Wu, Cheng [33] used a score of 19. In addition to the MMSE, the included studies used additional tests, such as NINCDS-ADRDA and clinical dementia ratings, to classify the types of dementia. Radiological examinations, such as CT scan and MRI, were used in three studies to facilitate the classification of dementia [20,28,35]. 

### 3.3. Quality of Included Studies

In general, the overall quality of the included studies was moderate. The samples in all studies were appropriate to represent the target population. The sample size was appropriate in most studies and was described in sufficient detail. The response rate was adequate in most studies. Two studies did not clearly state the sampling method. For example, Ji, Shi [11] selected samples from 56 out of 949 villages in rural areas, but the selection of these 56 villages was not described. The recruitment strategy used by Liao, Hunag [31] was not described. Several studies did not conduct subgroup analysis, and the resulting coverage of samples in different age groups was unclear [21,23,29,30,31,32]. Three studies did not clearly state the methods used for the identification of dementia [24,25,31], while the protocol of the standardized assessment of dementia remained unclear in four studies [19,24,25,31]. It should be emphasized that the appropriateness of statistical analysis was in doubt for six studies as there was no information about p-value or no detailed description of the statistical methods used [21,23,29,30,31,32].

### 3.4. Incidence, Prevalence, and Mortality of Dementia in China between 2010 and 2020

Three studies did not report the prevalence of dementia (Bo et al. 2019 [19] and Hu et al. 2018 [25] reported mortality, while Yang et al. 2016a [12] reported incidence). So, the meta-analysis of the prevalence of dementia was performed based on available data from 16 studies (Figure 2). The meta-analysis showed that the prevalence of dementia was 6% (95%CI 5%, 8%; I^2^ = 99%, *p* < 0.001). In particular, the subgroup analysis showed that the prevalence of AD was 5% (95%CI 4%, 6%; I^2^ = 76.6%, *p* = 0.014), and the prevalence of VaD was 1% (95%CI 0%, 2%; I^2^ = 91%, *p* = 0.001). The subgroup analysis of different areas (Figure 3) showed that the prevalence of dementia in rural areas was 6% (95%CI 4%, 8%; I^2^ = 87%, *p* < 0.001), the prevalence in urban areas was 6% (95%CI 4%, 8%; I^2^ = 98%, *p* < 0.001), and the prevalence reported in studies including both areas was 7% (95%CI 5%, 9%; I^2^ = 99.4%, *p* < 0.001). The subgroup analysis of years (Figure 4) showed that prevalence of dementia before 2015 was 6% (95%CI 5%, 8%; I^2^ = 98.5%, *p* < 0.001), and that after 2015 was 7% (95%CI 4%, 9%; I^2^ = 99.3%, *p* < 0.001). However, the results indicated substantial heterogeneity.

The incidence of dementia was reported in three studies (Table 2). Ding, Zhao [22] reported the incidence of dementia was 1.33% based on the Shanghai Epidemiological Survey of Dementia and AD in 2010 (n = 3670). Yu, Zhang [35] reported the incidence of dementia was 9.30% based on a survey of community retirees (n = 129). In addition to dementia, Yang, Chen [34] reported the incidence of AD was 13.13/1000 person-years.

Two studies reported mortality (Table 2). One study analyzed data at the national level and revealed that the number of people who died from dementia increased from 2845 to 12187 between 2010 and 2015 (n = 77 million) [19]. Within the same time frame, the mortality rate of dementia was reported to increase from 15.73 (1/10^5^) to 21.58 (1/10^5^) in another study (n = 778,389) which analyzed data at the city level [25].

### 3.5. Geographical, Age, and Sex Differences in the Prevalence and Incidence of Dementia in China between 2010 and 2020

As described in the above section, the prevalence and incidence of dementia did not show statistically significant differences in terms of geographical areas for rural and urban areas (Figure 3). There were insufficient data to conduct a meta-analysis based on age and sex at the same time, so we investigated these factors separately. Overall, ageing and being female were two risk factors for dementia. The prevalence and incidence of dementia increased with age, which increased sharply after the age of 80. After the age of 85, the prevalence of dementia could rise to over 22% [23,32]. Females had a higher prevalence and incidence of dementia compared to males. Ji, Shi [11] reported that the prevalence of dementia was twice as high in women as in men. Additionally, we found that the prevalence of dementia decreased by year in urban areas but not in rural areas (Figure 3), and the lowest prevalence of dementia was in the Qinghai Tibetan area (1.33%, 95%CI: 0.98–1.69) [21].

## 4. Discussion

In this systemic review, we included 19 studies conducted in mainland China from 2010 to 2020. A meta-analysis of data from 16 studies showed that the overall dementia prevalence was 6% in the population, while the pooled prevalence of AD and VaD was 5% and 1.0%, respectively. The prevalence rates of dementia and AD were greater than those reported in a previous review, which included studies published from 1985 to 2015 [37], but the prevalence of VaD was slightly lower than that reported in the previous review (1.09%). A recent review also found that the pooled prevalence of VaD had decreased slightly to 0.96% [38]. This may be related to chronic disease management in primary care settings in China, where hypertension amongst the older population has been better controlled, which potentially means that vascular diseases are detected earlier [39]. Compared to studies in other countries, the UK had a slightly higher overall dementia prevalence (7.2%) in 2019 [40]. In selected low- and-middle-income countries (Brazil, India, Indonesia, Jamaica, Kenya, Mexico, and South Africa), the prevalence ranged from 2% to 9% [41]. In addition, a systematic review on the prevalence of dementia in Europe reported that the prevalence rate after standardized for age and sex was 7.1% [42]. Other meta-analyses indicated that in Nigeria, the prevalence was 4.9% [43], whilst it was 2% in India [44] and 5.3% in South Korea [45]. This heterogeneity can be related to different factors of economic development, diagnostic criteria and rates, cultures, lifestyles, etc.

It is worth noting that one included study in this review reported that the prevalence of AD among Tibetans aged 60 years and older was only 1.33% [27]. There are many reasons behind this distinct gap. Tibetans have very different lifestyles, religious beliefs, and customs, as well as living at high altitudes, compared to other Chinese ethnic groups, and all of these factors may affect their health outcomes [46]. This is worth exploring further. Overall, the prevalence rates of dementia and AD in this review were similar to the rates reported in the World Alzheimer Report 2015 [47].

Our subgroup analysis found that there was no difference in the prevalence of dementia between people who lived in urban and rural areas (in both urban and rural areas, 6% of people had dementia). This is different from another review paper, which reported that the prevalence of dementia was significantly higher in rural areas than in urban areas [48]. One included study in our review also reported that the risk of dementia in rural areas was 1.558 times that of urban areas [21]. This decline in the gap of prevalence between urban and rural areas may be related to chronic disease management of older populations in rural communities and improvements in awareness of dementia services, as well as the integration of social medical insurance for rural residents, which have improved the level of early prevention and detection, and narrowed the unban–rural gap [49]. This rural–urban difference is in line with another finding of this review that dementia prevalence decreases by year in urban areas but not in rural areas. However, further research is needed to verify this change. In addition, economic development in urban and rural areas vary across the country. For example, the economy in some western urban regions may be less developed than in rural southern areas in China. This would affect local healthcare services or relevant support services, such as doctors’ expertise in dementia diagnosis, which would affect the prevalence of dementia. Another selected study reported that the age-standardized mortality in rural areas was higher than that in urban areas [19], and the findings of this individual study differed from our meta-analysis. The economic and healthcare service discrepancies between urban and rural areas may affect the mortality rate. The data on mortality in our review were not sufficient to conduct a meta-analysis. Further research using national data, especially in less developed northern and western regions, is needed. In addition, the subgroup analysis by year showed that the prevalence of dementia increased slightly from 6% (2010–2015) to 7% (2015–2020). This is consistent with Chan, Wang [9] who found that the prevalence of dementia increased with years.

In our review, Ding, Zhao [22], Wu, Cheng [33], Ji, Shi [11], and Jia, Du [28] reported that education was one of the most important factors influencing dementia prevalence. Education plays a protective role in the brain’s cognitive function [22,50], and people who have a higher education may be more aware of the risk factors related to dementia and be more likely to make reasonable decisions for their health management. The latter could be improved via educational interventions or training programs in the communities. Lower education as a dementia risk factor has been identified in many international studies as well [51,52], and this further indicates the importance of increasing public awareness of dementia. Engagement in social activities was identified as a protective factor against dementia in two included studies in this review [11,33]. This is in line with some international studies [53,54]. However, the methodology of these studies was observational and was combined with cognitive training and/or physical activities, rather than as a standardized interventional study, which means they lack strong evidence from which to draw any conclusions. So, more randomized controlled trials (RCTs) are needed in the future. 

Another interesting finding is that females have a higher prevalence and incidence of dementia compared to males in China, and the prevalence can be twice as high in women compared to men in some cases. This is consistent with another systematic review, which estimated the prevalence of dementia across four continents (Asia, Africa, Europe, and America); it reported that the number of females with dementia was greater than the number of males (788 cases versus 561 cases per 10,000 persons) [55].

Even though this study provided an updated estimate of the prevalence of dementia in China, potential limitations should be considered when interpreting and using the findings of our study. Although our study included national and regional data in China, the number of included participants was small for a country of 1.4 billion people. Aside from one national study, Huang, Shang [27] was the only study to include participants from less developed areas (Qinghai–Tibet Plateau). Data from other less developed areas were lacking. Given the vast expanse of China, distinct regions exhibit variations in demographic composition, environmental factors, and healthcare resources, and these result in divergent dementia incidence rates, which may cause bias in studies. Therefore, there needs to be separate subgroup analyses for populations from different regions to account for the potential impact of regional variations. Furthermore, different diagnostic criteria could contribute to the high heterogeneity between studies. The moderate quality of the included studies might lead to an over- or underestimation of the results of the meta-analysis. Because of the small number of included studies, the results from the subgroup analysis should be interpreted with caution. In addition, we were only able to collect educational information from four reviewed papers. So, we could not analyze this factor as this information was missing in other studies. But considering that education is an important risk factor of dementia, we present and discuss education in this discussion section. 

This review has implications for research and clinical practice. Applying consistent criteria to the diagnosis of dementia is necessary in future research to provide an accurate estimation of the prevalence of dementia in China. Furthermore, appropriate training for dementia specialists as well as rigorous quality control is essential for an accurate estimation. Moreover, early detection of dementia, especially in high-risk groups, is extremely important for clinical practice to prevent and alleviate the progression of dementia. However, public awareness of dementia is insufficient; a study showed that only 63.14% of Chinese people were aware of dementia or had some kind of knowledge on the topic [56]. Healthcare providers should cooperate with local communities to increase public knowledge and awareness of dementia to develop a dementia-friendly community. Training and support for dementia care are required since dementia is a progressive deterioration disorder. Mental burdens are increased in family caregivers of patients with dementia [28]. Community support and interventions, such as bibliotherapy or educational programs, could improve caregivers’ coping skills, psychological well-being, and attitude toward dementia [57].

## 5. Conclusions

This review found a large variation in dementia prevalence and incidence rates in China between 2010 and 2020. The prevalence and mortality of dementia increased year by year. A higher prevalence was observed in ageing and female populations, while no significant differences were detected in terms of geographic location. The results suggest that applying consistent criteria to the diagnosis of dementia is necessary in future research. Promotion of dementia knowledge and awareness at the community level is also needed. In addition, more studies are needed to investigate socio-demographic variables, such as educational factors.

## Figures and Tables

**Figure 1 healthcare-12-00334-f001:**
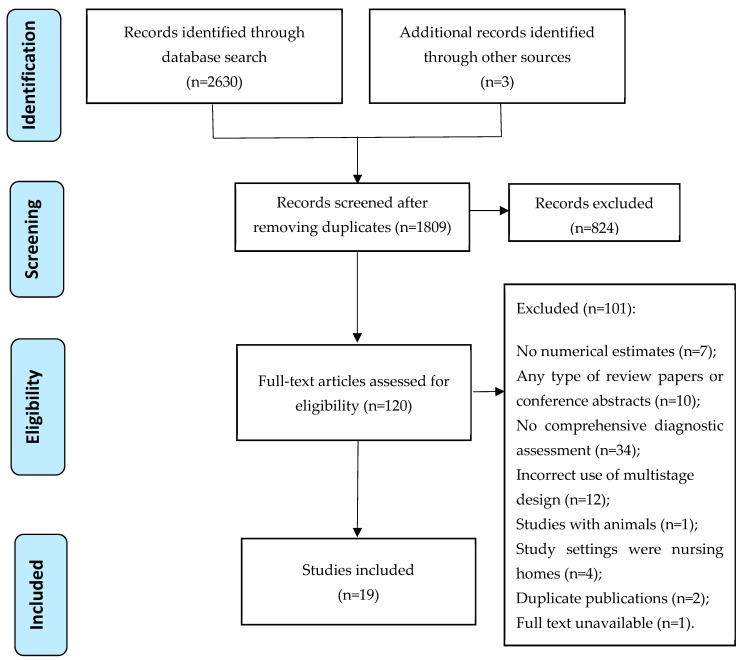
PRISMA flow chart showing the study screening process.

**Figure 2 healthcare-12-00334-f002:**
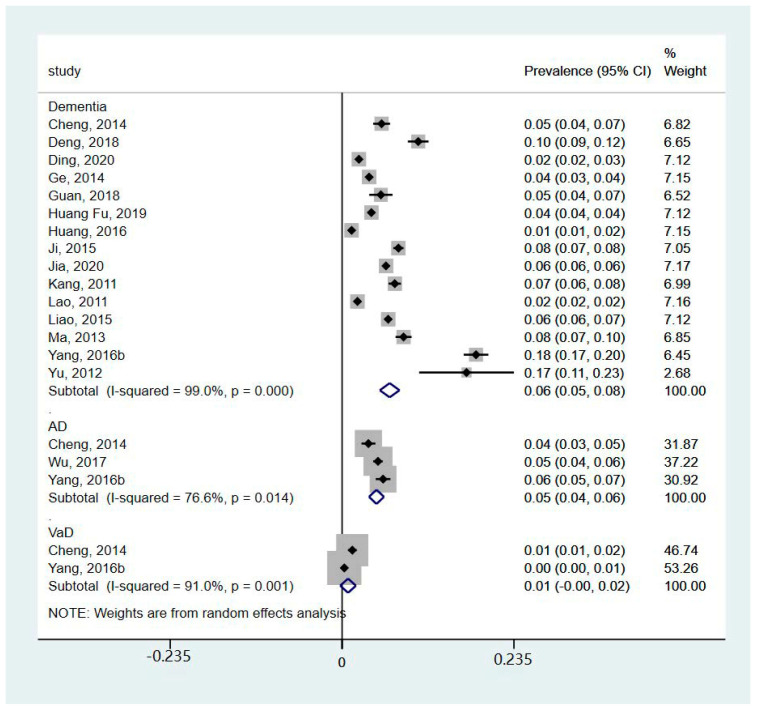
Forest plot of the prevalence of different types of dementia [11,20,21,22,23,24,26,27,28,29,30,31,32,33,34,35].

**Figure 3 healthcare-12-00334-f003:**
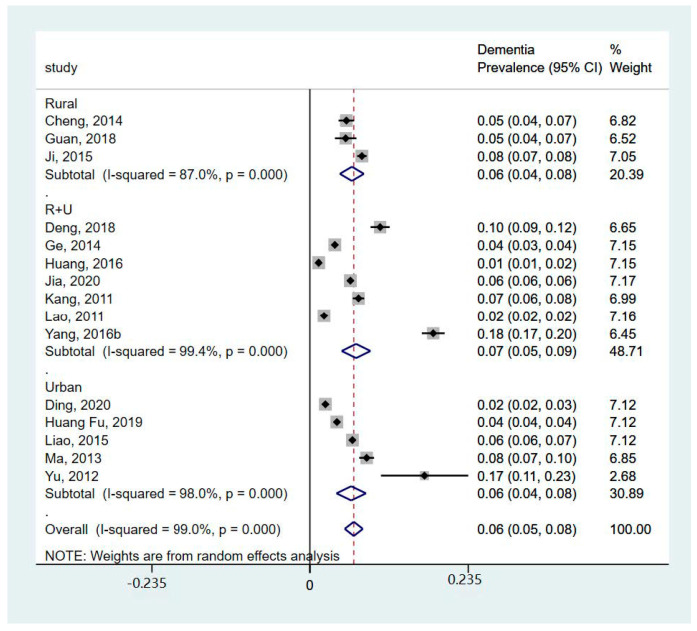
Forest plot of subgroup analysis of prevalence of dementia by area [11,20,21,22,23,24,26,27,28,29,30,31,32,34,35].

**Figure 4 healthcare-12-00334-f004:**
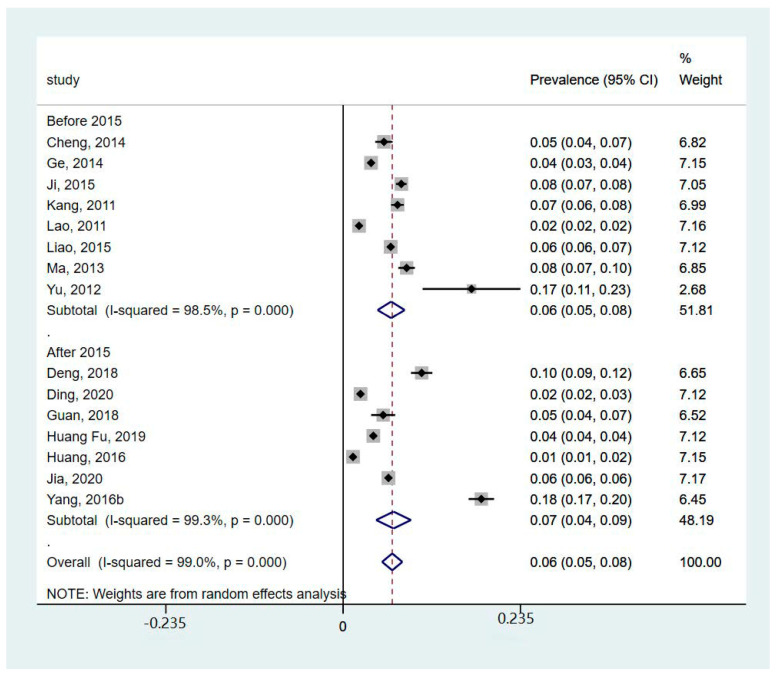
Forest plot of subgroup analysis of prevalence of dementia by year [11,20,21,22,23,24,26,27,28,29,30,31,32,34,35].

**Table 1 healthcare-12-00334-t001:** Critical appraisal of included studies.

Studies	Q1	Q2	Q3	Q4	Q5	Q6	Q7	Q8	Q9
Bo, 2019 [19]	Y	Y	Y	Y	Y	Y	U	Y	U
Cheng, 2014 [20]	Y	Y	Y	Y	Y	Y	Y	Y	Y
Deng, 2018 [21]	Y	Y	Y	Y	U	Y	Y	U	Y
Ding, 2020 [22]	Y	Y	Y	Y	Y	Y	Y	Y	Y
Ge, 2014 [23]	Y	Y	Y	Y	U	Y	Y	U	Y
Guan, 2018 [24]	Y	Y	Y	Y	Y	U	U	Y	Y
Hu, 2018 [25]	Y	N/A	Y	Y	Y	U	U	Y	N/A
Huang, 2019 [26]	Y	Y	Y	Y	Y	Y	Y	Y	Y
Huang, 2016 [27]	Y	Y	Y	Y	Y	Y	Y	Y	Y
Ji, 2015 [11]	Y	U	Y	Y	Y	Y	Y	Y	Y
Jia, 2020 [28]	Y	Y	Y	Y	Y	Y	Y	Y	Y
Kang, 2011 [29]	Y	Y	Y	Y	U	Y	Y	U	Y
Lao, 2011 [30]	Y	Y	Y	Y	U	Y	Y	U	Y
Liao, 2015 [31]	Y	U	Y	Y	U	U	U	U	Y
Ma, 2013 [32]	Y	Y	Y	Y	U	Y	Y	U	Y
Wu, 2017 [33]	Y	Y	Y	Y	Y	Y	Y	Y	Y
Yang, 2016a [12]	Y	Y	Y	Y	Y	Y	Y	Y	Y
Yang, 2016b [34]	Y	Y	Y	Y	Y	Y	Y	Y	Y
Yu, 2012 [35]	Y	Y	N	Y	Y	Y	Y	Y	Y

Notes: Y = yes; N = no; U = unclear; N/A = not applicable. JBI critical appraisal checklist for prevalence studies: Q1 = Was the sample frame appropriate to address the target population? Q2 = Were study participants sampled in an appropriate way? Q3 = Was the sample size adequate? Q4 = Were the study subjects and the setting described in detail? Q5 = Was the data analysis conducted with sufficient coverage of the identified sample? Q6 = Were valid methods used for the identification of the condition? Q7 = Was the condition measured in a standard, reliable way for all participants? Q8 = Was there appropriate statistical analysis? Q9 = Was the response rate adequate, and if not, was the low response rate managed appropriately?

**Table 2 healthcare-12-00334-t002:** Characteristics of included studies.

Studies	Design	Setting	Sample Size (Female)	Age	Case Identification	Prevalence	Incidence	Mortality
Bo, 2019 [19]	Longitudinal study	Both	77 million (49%)	All ages	ICD-10	N/A	N/A	2010 y: 2845; 2011 y: 2890; 2012 y: 3155; 2013 y: 10,654; 2014 y: 11,466; 2015 y: 12,187
Cheng, 2014 [20]	Cross-sectional study	Rural	1472 (54.8%)	69.7 ± 7.1	C-MMSE, DSM-IV, NIA-AA, NINDS-AIREN	Dementia: 5.37%; AD: 3.60%; VaD: 1.43%	N/A	N/A
Deng, 2018 [21]	Cross-sectional study	Both	1781 (60.5%)	60–69 y: 46% 70–79 y: 41.5% >80 y: 12.5%	C-MMSE, IADL	10.44%	N/A	N/A
Ding, 2020 [22]	Longitudinal study	Urban	3670 (55.4%)	55–64 y: 36% 65–74 y: 31% 75–84 y: 27.1% ≥85 y: 5.9%	C-MMSE, ADL, Clinical Dementia Rating Scale, DSM-IV	2.30%	1.33%	N/A
Ge, 2014 [23]	Cross-sectional study	Both	10,026 (51.7%)	75.4 ± 10.2	MMSE, DSM-IV	3.66%	N/A	N/A
Guan, 2018 [24]	Cross-sectional study	Rural	760 (47%)	60–69 y: 38.7% 70–79 y: 35% ≥80 y: 26.3%	GMS, self-developed dementia cognition scale	5.26%	N/A	N/A
Hu, 2018 [25]	Longitudinal study	Urban	778,389 (50.4%)	≥60	ICD-10	N/A	N/A	2010 y: 15.73 (1/10^5^) 2011 y: 13.94 (1/10^5^) 2012 y: 16.70 (1/10^5^) 2013 y: 22.81 (1/10^5^) 2014 y: 26.80 (1/10^5^) 2015 y: 21.58 (1/10^5^)
Huang Fu, 2019 [26]	Cross-sectional study	Urban	6419 (56.8%)	67.1 ± 5.4	DSM-IV, MMSE, ADL, MoCA, CCMD-3	4.02%	N/A	N/A
Huang, 2016 [27]	Cross-sectional study	Both	3974 (61.6%)	60–64 y: 24.5% 65–69 y: 24.8% 70–74 y: 20.9% 75–79 y: 16.4% 80–84 y: 9.2% ≥85 y: 4.2%	MMSE, ADL, Hachinski Ischemic Scale	1.33%	N/A	N/A
Ji, 2015 [11]	Cross-sectional study	Rural	5578 (55.5%)	≥60	MMSE	7.7%	N/A	N/A
Jia, 2020 [28]	Cross-sectional study	Both	46,011 (50.3%)	≥60	MMSE, MoCA, WHO California-Los Angeles Auditory Verbal test	6.0%	N/A	N/A
Kang, 2011 [29]	Cross-sectional study	Both	3632 (46.7%)	70.90 ± 7.2	MMSE	7.24%	N/A	N/A
Lao, 2011 [30]	Cross-sectional study	Both	7665 (54.2%)	66.4 ± 10.0	HDS	2.07%	N/A	N/A
Liao, 2015 [31]	Cross-sectional study	Urban	9733 (50.9%)	72.3 ± 6.5	MMSE	6.29%	N/A	N/A
Ma, 2013 [32]	Cross-sectional study	Urban	2442 (53.7%)	75.8 ± 7.6	MMSE	8.44%	N/A	N/A
Wu, 2017 [33]	Cross-sectional study	Both	4195 (51.1%)	71.0 ± 7.6	MMSE	AD: 4.89%	N/A	N/A
Yang, 2016a [12]	Longitudinal study	Urban	9733 (50.9%)	72.8± 7.8	MMSE	N/A	13.13/1000 person-years (AD)	N/A
Yang, 2016b [34]	Cross-sectional study	Both	2015 (58.8%)	79.5 ± 7.6	NIA-AA, MMSE, CDR	Dementia: 18.3%; AD: 5.6%; VaD: 0.3%	N/A	N/A
Yu, 2012 [35]	Longitudinal study	Urban	129 (34.1%)	83.5 ± 3.5	MMSE	17.02%	9.30%	N/A

Notes: ICD-10 = International Classification of Diseases, Tenth Revision; C-MMSE = The Chinese version of the Mini-Mental Status Examination; DSM = Diagnostic and Statistical Manual of Mental Disorders; NIA-AA = the National Institute on Aging and the Alzheimer’s Association; NINDS-AIREN = the National Institute of Neurological Disorders and Stroke/the Association Internationale pourla Recherche et l’Enseignement en Neurosciences; IADL = Instrumental Activities of Daily Living Scale; GMS = Geriatric Mental Status Scale; MoCA = Montreal Cognitive Assessment; CCMD-3 = Chinese Classification and Diagnostic Criteria of Mental Disorders; HDS = Hasegawa dementia scale; CDR = Clinical Dementia Rating Scale; AD= Alzheimer’s disease; VaD = vascular dementia; N/A = not applicable.

## Data Availability

Data are not obtained from a third party and are not publicly available. The full dataset and data analysis code following the receipt of ethics approval are available from the corresponding author.

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
