# Peer review of "Epidemiology of Dementia in China in 2010–2020: A Systematic Review and Meta-Analysis"

_healthcare, 2024, doi:10.3390/healthcare12030334_

Round 1

Reviewer 1 Report

Comments and Suggestions for Authors

This study described a meta-analysis to estimate the pooled prevalence of dementia from 19 studies from 2010 to 2020 in China. It showed a 6% prevalence with Alzheimer's disease at 5% and vascular dementia at 1%. They also found consistent rural and urban prevalence rates. The description of the study is clear basically. However, there are few significant points that need to address to improve.

First, the manuscript needs a thorough proofreading to enhance the clarity and coherence in its English expression. Also, please doublet checks the figures order and quality. The order of the figures are messed up.

In 2.2, what are the “internationally recognized definitions of Alzheimer's disease (AD) 95 and other types of dementia;” referred here? And for the “the incorrect use of multistage design”, can the authors elaborate more on this?

In the data analysis, it’s mentioned that the significance of heterogeneity from chi-square test was defined as p<0.1, which is not a conventional threshold for statistical significance. How do authors justify this?

 Suggest providing a bit more detail on how the qualitative analysis will be conducted. What factors will be considered in the synthesis?

Figure1: please remove the paragraph symbol.

3.3. Quality of Included Studies – line 180- the description is vague. If the appropriateness of the statistical analysis in certain studies is in doubt, it would be more informative to provide specific reasons or concerns. This adds clarity to your critique.

In 3.4, the studies decreased from 19 to16. Which three were excluded and why?

Table1, study for Bo,2019, is there a reported age range for the all the included people?

In figure 3, although there is no significant differences between rural and urban areas, the dementia prevalence was decreased by time in urban area but not in rural. Suggest the authors to analysis these factors integrally in addition to investigate them one by one. Also, for the area difference, did the authors compare between eastern, central, and western areas?

Comments on the Quality of English Language

The manuscript needs a thorough proofreading to enhance the clarity and coherence in its English expression.

Author Response

Dear reviewer 1,

We greatly appreciate the comments you provided, which have helped us to improve the quality of the paper. We have considered all your comments and carefully revised our paper. We highlighted the changes to the manuscript with track changes. Thank you again for your kind reviews.

Reviewer 1 Comments and Suggestions for Authors

First, the manuscript needs a thorough proofreading to enhance the clarity and coherence in its English expression. Also, please doublet checks the figures order and quality. The order of the figures are messed up.

Authors’ response: Many thanks for your comments. We have checked and proofread the manuscript. We also checked the order of the figures and revised where needed.

In 2.2, what are the “internationally recognized definitions of Alzheimer's disease (AD) and other types of dementia;” referred here? And for the “the incorrect use of multistage design”, can the authors elaborate more on this?

Authors response: Many thanks for your comments. ‘The internationally recognized definitions of Alzheimer's disease (AD) and other types of dementia’ referred as: for example, ICD-10, C-MMSE etc. please see the detailed notes under the bottom of Table 2.

‘The incorrect use of multistage design’ refers to, e.g., the sample size was too small or only the individual circumstances of specific subgroups or cases were concerned. We have elaborated this further in the manuscript (Section 2.2 Eligible criteria—exclusion criteria).

In the data analysis, it’s mentioned that the significance of heterogeneity from chi-square test was defined as p<0.1, which is not a conventional threshold for statistical significance. How do authors justify this?

Authors response: According to the Cochrane handbook and Higgins (2003), in meta-analysis, the chi-square test is used to estimate the heterogeneity and the p-value can be set as <0.1.

Higgins, J.P.T., Thompson, S.G., Deeks, J.J., Altman, D.G., 2003. Measuring inconsistency in meta-analyses. (Education and debate). BMJ. 327 (7414), 557.

Suggest providing a bit more detail on how the qualitative analysis will be conducted. What factors will be considered in the synthesis?

Authors response: Sorry for the unclear clarity of the language and confusion, there was no qualitative analysis. We have revised the last sentence in 2.8 Data Analysis.

Figure1: please remove the paragraph symbol.

Authors response: We have checked and removed the paragraph symbol.

3.3. Quality of Included Studies – line 180- the description is vague. If the appropriateness of the statistical analysis in certain studies is in doubt, it would be more informative to provide specific reasons or concerns. This adds clarity to your critique.

Authors response: We revised the sentence into the following:

It should be emphasized that the appropriateness of statistical analysis was in doubt in six studies as there was no information about p-values or no detailed descriptions of statistical method. Please see 3.3 Quality of Included Studies, line 188-189

In 3.4, the studies decreased from 19 to16. Which three were excluded and why?

Authors response: In line with Table 2, Bo et al. (2019), Hu et al. (2018), and Yang et al. (2016a) were the studies excluded from analysis since the prevalence of dementia was not reported. We have added further explanation in 3.4, line 191-192.

Table1, study for Bo,2019, is there a reported age range for the all the included people?

Authors response: We read Bo et al., (2019) study again but no age or age range was specified.

In figure 3, although there is no significant differences between rural and urban areas, the dementia prevalence was decreased by time in urban area but not in rural. Suggest the authors to analysis these factors integrally in addition to investigate them one by one. Also, for the area difference, did the authors compare between eastern, central, and western areas?

Authors response: We have followed the reviewer’s suggestion to analysis these factors integrally in addition to investigate them one by one and reported as: “Additionally, we found the prevalence of dementia decreased by year in urban area but not in rural area (Fig. 3), and the lowest prevalence of dementia was in Qinghai Tibetan area (1.33%, 95%CI: 0.98-1.69). Please see section 3.5. line 233-236 and Discussion part Line 272-273.

Comments on the Quality of English Language

The manuscript needs a thorough proofreading to enhance the clarity and coherence in its English expression.

Authors response: We have checked and proofread the manuscript.

Reviewer 2 Report

Comments and Suggestions for Authors

This systematic review explored the prevalence and incidence of dementia in China, accounting for any geographical, age, or gender difference in these rates. It is a well-written paper that uses a clear and robust methodology, and I only have a few minor comments.

- If it is possible, when you cite the reference in the text, report the first author and “et al.” (instead of reporting the names of the two first authors).

- In the introduction, I would explain better why you chose these specific factors (geographical area, age, and gender). This could help the non-expert reader.

- In the discussion, I did not report the comment on education [lines 261-274], or you could explain why you did not collect and analyse this information in this review. In the conclusion, you could report the need for more studies investigating this socio-demographic variable.

- As you started to do in lines 235-237, I would like to see more comparisons between the data you found for China and other countries (high, middle, and low-income countries). This could be a good section to compare different prevalences/incidences in different cultures.

Author Response

Dear reviewer 2,

We greatly appreciate the comments you provided, which have helped us to improve the quality of the paper. We have considered all your comments and carefully revised our paper. We highlighted the changes to the manuscript with track changes. Thank you again for your kind reviews.

Reviewer 2 Comments and Suggestions for Authors

This systematic review explored the prevalence and incidence of dementia in China, accounting for any geographical, age, or gender difference in these rates. It is a well-written paper that uses a clear and robust methodology, and I only have a few minor comments.

- If it is possible, when you cite the reference in the text, report the first author and “et al.” (instead of reporting the names of the two first authors).

Authors response: Many thanks for your comments. We have checked and revised the references in the text.

- In the introduction, I would explain better why you chose these specific factors (geographical area, age, and gender). This could help the non-expert reader.

Authors response: According to your suggestions. We have further explained why we chose these specific factors (geographical area, age, and gender) and added the sentence ‘In addition, due to the large diversity of the healthcare services, customs and lifestyles across the country, the differences in prevalence by age, sex and geography were also investigated in order to inform related stakeholders. For example, previous study indicates that there was a greater inequity of outpatient service use in urban areas than that in rural areas in China’ in the introduction section, please see the page 2, line 78-82.

- In the discussion, I did not report the comment on education [lines 261-274], or you could explain why you did not collect and analyse this information in this review. In the conclusion, you could report the need for more studies investigating this socio-demographic variable.

Authors response: Many thanks for your comments. We explain why we did not analyse the information on education in this review and added: ‘In addition, we were only able to collect information on education from four reviewed papers. So, we cannot analyse this factor as this dataset is missing from other studies, but consideration for education is an important risk factor of dementia, so we presented and discussed the education factor in this discussion section.’ Please see page 14, line 323-327.

In the conclusion, we added: ‘In addition, more studies are needed to investigate the socio-demographic variable, such as education.’ Please see page14, line 350-351.

- As you started to do in lines 235-237, I would like to see more comparisons between the data you found for China and other countries (high, middle, and low-income countries). This could be a good section to compare different prevalences/incidences in different cultures.

Authors response: Many thanks for your comments. comparisons between the data in China and other countries has been added as: ‘Compared to studies in other countries, the UK has a slightly higher overall dementia prevalence (7.2%) in 2019 [40], in the selected low-and-middle income countries (Brazil, India, Indonesia, Jamaica, Kenya, Mexico, and South Africa), the prevalence ranged from 2% to 9% [41]. From these reported studies, it is difficult to give a conclusion on whether economic development of a country is associated with dementia prevalence.’ Please see page 13, line 249-254.

Reviewer 3 Report

Comments and Suggestions for Authors

The submitted manuscript explores the incidence, prevalence, and mortality of dementia in China between 2010 and 2020, shedding light on potential age, and sex differences in the prevalence and incidence of dementia. Overall, the manuscript provides valuable insights into the epidemiology of dementia in China. Here are some concerns about the article:

1.There are inconsistencies in Figure 1. In the screening phase, the authors mention the removal of duplicate papers. However, in the eligibility phase, they again state the exclusion of two duplicate papers. It is recommended to consolidate information regarding duplicate papers into a unified presentation.

2.Given the vast expanse of China, distinct regions exhibit variations in demographic composition, environmental factors, and healthcare resources. These disparities may result in divergent Dementia incidence rates and etiologies. Consequently, the authors must consider whether the studies included constitute a comprehensive investigation of China as a whole or if they are regionally focused. It is imperative to conduct separate subgroup analyses for populations from different regions to account for the potential impact of regional variations on the study outcomes.

3.The authors include 19 articles in the study. The screening process does not provide clarification on whether the utilized study populations are mutually independent. The potential non-independence of these sampled populations across studies will introduce the risk of bias. In the meta-analysis report, it is imperative for the authors to explicitly address the presence or absence of repeated use of subject cohorts and elucidate the methodology employed to address this issue. 

4.The article underscores the pivotal role of education as one of the most influential factors affecting dementia prevalence. Education serves as a critical determinant in constructing cognitive reserve, and a substantial body of post-mortem examinations reveals that many individuals who were not clinically diagnosed with dementia during their lifetime exhibit distinct dementia-related features upon autopsy. This phenomenon is attributed to the protective effect of cognitive reserve, which delays the onset of dementia. Given that a majority of studies involved in the author's research rely on dementia diagnoses based on cognitive assessments, it becomes imperative to engage in a thorough discussion regarding the potential impact of cognitive reserve (educational attainment).

5.The author should undertake a comparative analysis between the findings presented in the article and the broader epidemiological landscape of dementia worldwide. Such a comparative exploration is essential for discerning the similarities and divergences in the epidemiology of dementia across different countries.

Author Response

Dear reviewer 3,

We greatly appreciate the comments you provided, which have helped us to improve the quality of the paper. We have considered all your comments and carefully revised our paper. We highlighted the changes to the manuscript with track changes. Thank you again for your kind reviews.

Reviewer 3 Comments and Suggestions for Authors

  1. There are inconsistencies in Figure 1. In the screening phase, the authors mention the removal of duplicate papers. However, in the eligibility phase, they again state the exclusion of two duplicate papers. It is recommended to consolidate information regarding duplicate papers into a unified presentation.

Authors’ response: Many thanks for your comments. The exclusion of another two duplicate papers is because they are different manuscripts but yield from the same study. We have revised the description of “duplicate publication” into “duplicate study reports” to make it clear.

2.Given the vast expanse of China, distinct regions exhibit variations in demographic composition, environmental factors, and healthcare resources. These disparities may result in divergent Dementia incidence rates and etiologies. Consequently, the authors must consider whether the studies included constitute a comprehensive investigation of China as a whole or if they are regionally focused. It is imperative to conduct separate subgroup analyses for populations from different regions to account for the potential impact of regional variations on the study outcomes.

Authors’ response: We agree with your comments and agree that this is one of the limitations of the study, so we added: ‘Given the vast expanse of China, distinct regions exhibit variations in demographic composition, environmental factors, and healthcare resources, these result in divergent dementia incidence rates which may cause the study bias. Therefore, there is a need for conducting separate subgroup analyses for populations from different regions, accounting for the potential impact of regional variations.’ Please see page 14, line 315-319.

3.The authors include 19 articles in the study. The screening process does not provide clarification on whether the utilized study populations are mutually independent. The potential non-independence of these sampled populations across studies will introduce the risk of bias. In the meta-analysis report, it is imperative for the authors to explicitly address the presence or absence of repeated use of subject cohorts and elucidate the methodology employed to address this issue.

Authors’ response: Many thanks for your comments.

Except for the national registry used in Bo et al., (2019) study, rest of the studies were conducted in different places that should be considered as mutually independent. Since Bo et al,(2019) was not included in the meta-analysis, our findings should not be influenced.

4.The article underscores the pivotal role of education as one of the most influential factors affecting dementia prevalence. Education serves as a critical determinant in constructing cognitive reserve, and a substantial body of post-mortem examinations reveals that many individuals who were not clinically diagnosed with dementia during their lifetime exhibit distinct dementia-related features upon autopsy. This phenomenon is attributed to the protective effect of cognitive reserve, which delays the onset of dementia. Given that a majority of studies involved in the author's research rely on dementia diagnoses based on cognitive assessments, it becomes imperative to engage in a thorough discussion regarding the potential impact of cognitive reserve (educational attainment).

Authors’ response: We agree with your comments, however we were only able to collect information on education from four reviewed papers. So, we could not analyse this factor as this dataset is missing from other studies, but consideration for education is an important risk factor of dementia, so we presented and discussed the factor of education in the discussion section. Please see discussion section, page 13, line 288-302. We also added this as one of the limitations of the study, please see page 123 line 323-327.

5.The author should undertake a comparative analysis between the findings presented in the article and the broader epidemiological landscape of dementia worldwide. Such a comparative exploration is essential for discerning the similarities and divergences in the epidemiology of dementia across different countries.

Authors’ response: Many thanks for the suggestion. Currently, we cannot undertake a comparative analysis between the findings presented in the article and the broader epidemiological landscape of dementia worldwide. As we only included studies in China, we did not collect epidemiological data of other countries. However, we did a comparison discussion of dementia prevalence within high, low-and-middle income countries in the discussion section. Please see page 12 line 249-254.

Round 2

Reviewer 1 Report

Comments and Suggestions for Authors

My questions have been solved.

Author Response

Dear Review 1,

Many thanks for your second review. 

Reviewer 3 Report

Comments and Suggestions for Authors

The comparison raised in the previous query does not entail an analysis of disparate national datasets. Instead, it pertains to the abundance of existing meta-analyses on the epidemiology of dementia. It is requested that the findings of the current study be compared with those of similar investigations in other countries.

Author Response

Dear reviewer 3,

We greatly appreciate the comments you provided, which have helped us to improve the quality of the paper. We have considered all your comments and carefully revised our paper. We highlighted the changes to the manuscript with track changes. Thank you again for your kind reviews.

Authors’ response: Sorry for the incorrect response in the previous query, we added a comparison with similar investigations in other countries in the discussion section, line 243-248. Thanks again for your comment.